# “Losing Faith in My Body”: Body Image in Individuals Diagnosed with End-Stage Renal Disease as Reflected in Drawings and Narratives

**DOI:** 10.3390/ijerph191710777

**Published:** 2022-08-30

**Authors:** Rachel Lev-Wiesel, Liraz Sasson, Netta Scharf, Yasmeen Abu Saleh, Anat Glikman, Denis Hazan, Yarden Shacham, Keren Barak-Doenyas

**Affiliations:** 1Social Work Department, Tel Hai College, Qiryat Shemona 1220800, Israel; 2The Sagol Center for Hyperbaric Medical Treatment and Research, Shamir Medical Center, Be’er Ya’akov 7035000, Israel

**Keywords:** end-stage renal disease, dialysis, patients, drawings, narratives

## Abstract

Chronic kidney disease (CKD) and the dependency on dialysis is an abrupt life-changing event that harms a patient’s life (e.g., social relationships, work, and well-being). This study aimed to examine how individuals who undergo chronic dialysis due to failure end-stage renal disease perceive their bodies, as reflected in drawings and narratives. Following ethical approval and signing a consent form to participate in the study, 29 adults between the ages of 20 and 85 who have undergone dialysis filled out an anonymous questionnaire that consisted of the following measures: The Center for Epidemiological Studies—Depression (CES-D), The Multidimensional Body-Self Relations Questionnaire (MBSRQ), and The MOS 36-Item Short-Form Health Survey (SF-36). After completion, they were asked to draw their self-figure before and after being diagnosed and narrate it. The data were quantitatively and narratively analyzed. The results revealed high levels of depression and concerns regarding body fitness and weight. Few significant differences were noted between self-figured drawings before and after the diagnosis, such as the body line and gender markers. Additionally, Fitness Evaluation and Overweight Preoccupation were significant among the participants.

## 1. Introduction

End-stage renal disease (ESRD) is associated with an increased morbidity risk, and mortality rates affecting around 14.5% of the world population [1]. Patients with ESRD experience psychological distress, notably depression and anxiety, compared with those with other chronic illnesses [2]. There are additional bio-physiological and psychosocial stressors that may contribute to the level of patient’s stress: dependency on the life-sustaining treatment of dialysis [3], neuromuscular, gastrointestinal, sexual dysfunction, chronic pain, fatigue, sleep disturbances [4,5], dietary restrictions, time constraints, and fear of death [6]. CKD also affects employment, relationships, and lifestyle [7]. Ref. [8] found that most patients find the transition to dialysis frightening and traumatic, in addition to the dialysis procedure itself, which contributes to the patients’ distress.

Ref. [9] claimed that people’s bodies affect how they think about themselves. Becoming ill unexpectedly with a chronic disease usually shakes the person’s life schemes, forcing the person to adjust abruptly to a new bodily self, moving from illusory ownership of a strong, healthy body to an ill, weak one that is dependent on others for survival [10]. The person might perceive the transition from a healthy body to a helpless, suffering body as a body betrayal [11], thus modifying their beliefs about the self [12]. This, in turn, may contribute to distress and harm the ability to cope with such a situation [11].

The effect of chronic illness on changes in body image and self-perception has been explored in studies on various chronic diseases such as breast cancer [13], fibromyalgia [14], and diabetes [15]. However, the impact of ESRD on a person’s body image and self-perception, to the best of our knowledge, has remained relatively unexplored. Indeed, several researchers highlight the need for further investigation of psychological distress in ESRD patients [1,3,16].

The purpose of the current retrospective study was to inquire how people whose life had changed due to ESRD perceive their bodies. Specifically, the study sought to investigate ESRD patients’ body image before the diagnosis of renal failure and at present, as reflected in self-figured drawings and narratives. In addition, we sought to give a voice to CKD patients with a creative tool.

## 2. Background

### 2.1. ESRD and Dialysis Medical Procedure

ESRD is a disease that causes permanent loss of renal function. The syndrome that results from the loss of kidney function and the accumulation of waste materials usually excreted by the kidney is called uremia [17]. Uremia is characterized clinically by loss of appetite, nausea, vomiting, insomnia, cognitive changes, and eventually coma [18]. When the residual renal function cannot supply the body’s needs, renal replacement therapy is needed. Kidney transplantation will be the treatment of choice for young and relatively healthy patients, but when a patient’s characteristics or lack of kidney donor excludes transplantation, hemodialysis or peritoneal dialysis is given. Hemodialysis protocol usually consists of treatment that is three or four hours long, given weekly sessions in the dialysis unit, while peritoneal dialysis is performed at home by the patients or caregivers [19]. Even under proper dialysis treatment, ESRD is associated with some degree of uremia and obligates tight restriction of fluids and electrolytes intake. Frequent infections increase cardiovascular risk, and sharp hemodynamic changes negatively affect dialysis patients’ quality of life and life expectancy [20].

### 2.2. ESRD and Dialysis Procedure as a Continuous Stressful Situation

ESRD lasts throughout the patient’s lifetime and affects all aspects of life, losing their health and independence and being obligated to dialysis treatment three or four times a week. The disease limits normal functioning [21] and dependency on the life-sustaining treatment of dialysis [3].

Numerous psychosocial stressors affect patients with ESRD. These include effects of diseases and treatment, functional limitations, sexual dysfunction, dietary restrictions, time constraints, and fear of death [22]. These stressors contribute to the expected psychological comorbidities, including depression, the most common diagnosis, affecting 10–50% of dialysis patients [6]. The dialysis procedure is a continuous stressful situation that patients find frightening and traumatic [8].

### 2.3. Body Image and Self-Perception

Self-perception is a person’s view of him/herself or any mental or physical attributes that constitute the self [23]. Body image is defined as one’s body-related self-perceptions and self-attitudes, including body-related thoughts, feelings, and behaviors [24,25,26]. The concept of body image includes two components: perceptual body image (i.e., how people perceive the size, weight, and shape of their body) and attitudinal body image (affective, cognitive, and behavioral concerns with one’s body size, weight, and shape) [27]. Self-rated body image (SRBI) is a psychological construct that refers to an individual’s self-perception of his/her body, linked with self-image and feelings [28]. The research found that individuals with a distorted body image tend to suffer from low self-concept, self-unworthiness, and higher levels of stress and depression [29,30]. For example, a study examining manifestations of distress and changes in self-perception in women diagnosed with breast cancer found that changes in self-perception occurred in women with the disease that may indicate a sense of femininity impairment and loss of attractiveness [13]. The question concerning the impact of ESRD on a person’s body image and self-perception remained, therefore, to be explored. To the best of our knowledge, despite its prevalence, there is a lack of studies focusing on patients with renal failure dependent on dialysis.

### 2.4. Self-Figure Drawing as a Tool for Personal Well-Being

Drawing oneself or drawing a human figure is a frequently used projective technique for psychological assessment [31]. Healthcare providers use the self-figured drawing analysis for evaluation and therapeutic purposes [32].

In recent years, there have been developments in the field of “Draw a Person” (DAP) based on assessments of specific population groups. The DAP effectively differentiated violent and non-violent male offenders, a finding that could facilitate the detection of violent tendencies among incarcerated males [33]. The DAP also differentiated between diagnosed schizophrenics and non-schizophrenics [34]. Self-figure drawings among colorectal cancer patients undergoing stoma surgery indicated psychological distress and a threat to physical and mental well-being [35]. Recently, [36] studied the ability to utilize the DAP to detect eating disorders with promising results [36]. Based on the latter, we used self-figure drawings as an assessment tool among ESRD patients in the current study.

### 2.5. Research Questions and Hypotheses

Based on the above review, (1) ESRD is a disease that negatively affects the person’s psychological and physical well-being [6,8,22], (2) body image is associated with self-perception [24,25,26], and (3) self-figure drawing can be used to assess self-perception and well-being [32]. The main research question was how individuals who undergo chronic dialysis due to ESRD perceive their bodies before (retrospectively) and currently as reflected in drawings and narratives.

We made the following hypotheses: (1) Self-figure drawings before and after the diagnosis will differ in the following indicators: body line (shaking, disconnected after illness; feeling of helplessness and fatigue), face impression (sadder after diagnosis), standing (fierce before the disease), and arms and hands (connected before illness, disconnected after). (2) A correlation between the variables body image, depression, quality of life, and drawing indicators will be found.

## 3. Method

### 3.1. Participants and Procedure

A convenience sample (*n* = 29, age ranges from 47 to 78, Mean = 66.3) of people who were diagnosed with ESRD and treated by chronic hemodialysis or peritoneal dialysis (from 1 year to 27 years, median = 3 years) in the Nephrology and Dialysis unit of Shamir Medical Center (Assaf Harofe) participated in this retrospective study. Patients who suffered from additional chronic illnesses, such as cancer, were excluded. Table 1 presents the distribution of all demographic variables.

Following ethical approval by The Shamir Helsinki Committee (0246-21-ASF) and signing on informed consent, a self-report questionnaire was administered (in a digital form Qualtrics). It consisted of the following measures: demographic characteristics (e.g., age and gender), depression (Center for Epidemiologic Studies Depression Scale CES-D), the Multidimensional Body-Self-Relations Questionnaire (MBSRQ), and the quality of life (SF-36 Short Form). During the four hours of dialysis, participants filled out the questionnaire and were given two A4 (21 × 29.7 cm) sheets of paper and a pencil at completion. They were asked to draw their self-figure before being diagnosed with renal failure and after their diagnosis (at present), with no time limit, and then to provide a narrative.

### 3.2. Measures

A mix-methods approach was employed: qualitative measures included drawings and narratives; the quantitative measures included demographics, depression scale (CES-D), body perception (MBSRQ), and quality of life (SF-36). All the measures were adapted to Hebrew and extensively used in the Hebrew-speaking population (e.g., [37]).

The Center for Epidemiological Studies—Depression (CES-D), originally published by Radloff in 1977, is a 20-item designed to identify depression among the general population. The items on this scale primarily measure affective and somatic aspects of depression, such as restless sleep (i.e., “My sleep was restless”), poor appetite (i.e., “I did not feel like eating; my appetite was poor”), and feeling lonely (i.e., “I felt lonely”). Response options range from 0 to 3 for each item (0 = Rarely or None of the Time, 1 = Some or Little of the Time, two = Moderately or Much of the time, three = Most or Almost All the Time) (chief psychologist, n.d.). Scores range from 0 to 60, with high scores indicating more significant depressive symptoms. Items nos. 4, 8, 12, and 16 are inverted items (so they have to be reversed, i.e., to reduce the score from 3). The higher the overall score, the more it indicates that the person reports suffering from depressive symptoms. A score of 1–10 indicates low depressive symptoms, a score of 11–16 indicates mild depressive symptoms, and a score of 17 or higher indicates high-grade depressive symptoms (chief psychologist, n.d.).

The CES-D provides cutoff scores (e.g., 16 or greater) that evaluate the risk for clinical depression with good sensitivity, specificity, and high internal consistency. Four specific factors are generally described as positive effects, depressed or negative effects, somatic symptoms, and interpersonal problems [38]. The questionnaire has good psychometric data, high internal reliability (Alpha Cronbach index = 85.0 in the general population and 9.0 in the clinical population), and high validity indices. High correlations between self-report questionnaire scores for depression and patients’ depression levels are rated by clinicians [38]. High psychometric indices were also found in Israeli society [39,40].

The Multidimensional Body-Self Relations Questionnaire (MBSRQ) [41] evaluates different attitudinal facets of body image [42]. It consists of a 69-item divided into seven-factor subscales reflecting two dispositional dimensions—“Evaluation” and cognitive-behavioral “Orientation”—vis-à-vis each of the three bodily domains of “Appearance” (i.e., “Before I go out in a public place, I always pay attention to what I look like”), “Fitness” (i.e., “I am very aware of small changes in my weight”), and “Health/Illness” (i.e., “I am very aware of small changes in my physical health”). In addition to its seven Factor Subscales, the MBSRQ has three special multi-item subscales: (1) The Body Areas Satisfaction Scale (BASS) approaches body-image evaluation as dissatisfaction–satisfaction with body areas and attributes. (2) The Overweight Preoccupation Scale assesses fat anxiety, weight vigilance, dieting, and eating restraint. (3) The Self-Classified Weight Scale assesses self-appraisals of weight from “very underweight” to “very overweight”. Items’ responses range from 1 to 5; the highest score is a better state.

Appearance Evaluation includes seven items. Appearance Orientation consists of 12 items, Fitness Evaluation 3 items, Fitness Orientation comprises 13 items, Health Evaluation consists of 6 items, Health Orientation includes eight items, Illness Orientation contains 5 items, The Body Areas Satisfaction Scale (BASS) includes 9 items, and The Overweight Preoccupation has 4 items. The Self-Classified Weight consists of 2 items [18,43]. Subscale scores are the means of the constituent items after reversing contraindication items.

Subscale Interpretation. Appearance Evaluation (AE): Feelings of physical attractiveness or unattractiveness; satisfaction or dissatisfaction with one’s looks. High scorers feel primarily positive and satisfied with their appearance; low scorers have general unhappiness with their physical appearance [42]. Appearance Orientation (AO): Extent of investment in one’s appearance. High scorers place more importance on how they look, pay attention to their appearance, and engage in extensive grooming behaviors. Low scorers are apathetic about their appearance; their looks are not especially important, and they do not expend much effort to “look good”. Fitness Evaluation (FE): Feelings of being physically fit or unfit. High scorers regard themselves as physically fit, “in shape”, or athletically active and competent. Low scorers feel physically unfit, “out of shape”, or athletically unskilled. High scorers value fitness and actively engage in activities to enhance or maintain their fitness. Low scorers do not value physical fitness and do not regularly incorporate exercise activities into their lifestyles [42]. Fitness Orientation (FO): Extent of investment in being physically fit or athletically competent. High scorers value fitness and actively engage in activities to enhance or maintain their fitness. Low scorers do not value physical fitness and do not regularly incorporate exercise activities into their lifestyles. Health Evaluation (HE): Feelings of physical health and the freedom from physical illness. High scorers feel their bodies are in good health. Low scorers feel unhealthy and experience bodily symptoms of disease or vulnerability to illness. Health Orientation (HO): Extent of investment in a physically healthy lifestyle. High scorers are health-aware individuals who try to lead a healthy lifestyle. Low scorers are more apathetic about their health [42]. Illness Orientation (IO): Extent of reactivity to being or becoming ill. High scorers are alert to symptoms of physical illness and are apt to seek medical attention. Low scorers are not especially alert or reactive to the physical signs of disease. Body Areas Satisfaction Scale (BASS): Similar to the Appearance Evaluation subscale, except that the BASS taps satisfaction with discrete aspects of one’s appearance. High composite scorers are generally content with most areas of their bodies. Low scorers are unhappy with the size or appearance of several areas [42]. Overweight Preoccupation (OWP): This scale assesses a construct reflecting fat anxiety, weight vigilance, dieting, and eating restraint. Self-Classified Weight (SCW): This scale reflects how one perceives and labels one’s weight, from very underweight to very overweight.

Among males, alphas range from 0.70 for SCW scores to 0.91 for FO scores, and among females, alphas range from 0.73 (BASS) to 0.90 (FO). The test–retest stability of the MBSRQ subscale scores is also acceptable over one month. Among males, the coefficients range from 0.71 (HE) to 0.89 (AO), and among females, the coefficients range from 0.74 (SCW and BASS) to 0.91 (AE) [42]. In our study, the following alphas were calculated: SCW, 0.83; FO, 0.76; BASS, 0.87; HE, 0.54; AO, 0.80; and AE, 0.78.

The MOS 36-Item Short-Form Health Survey (SF-36), developed by [43], includes 36 items that describe eight multi-item health dimensions, covering functional status, well-being, and overall evaluation of health. The first dimension, which measures functional status, includes Physical Functioning (PF)—10 items (i.e., “Bathing or wearing clothes without help”); Social Functioning (SF)—2 items (i.e., “In the last four weeks, to what extent have your physical health condition or your emotional problems to your usual social activities with family, friends, neighbors?”); Role Limitations due to Physical Health Problems (RP)—4 items (i.e., “Were you limited in performing any work or other activities?”); and Role Limitations due to Personal or Emotional Problems (RE)—3 items (i.e., “Have you reduced the amount of time you have devoted to your work or other activities”). The second dimension measuring well-being includes Bodily Pain (BP)—2 items (i.e., “To what extent have you had physical pain in the last four weeks?”); General Mental Health (MH)—5 items (i.e., “You were very nervous”); Vitality (Fatigue/Energy) (VIT)—4 items (i.e., “Did you feel full of energy”); and the third dimension measuring overall evaluation of health include General Health Perceptions (GH)—5 items (i.e., “How would you rate your health in general”), and one item measuring reported health transition [44]. The response range was from 1 to 5; a higher score represents better health. Reliability exceeded the minimum standard of 0.70 recommended for measures used in in-group comparisons in more than 25 studies; most have exceeded 0.80. Reliability for physical and mental summary scores exceeded 0.90. The median reliability coefficient for each of the eight scales is equal to or greater than 0.80 except for Social Functioning, which had median reliability across studies of 0.76 [44]. Our study’s alpha coefficients range from 0.68 (GH) to 0.92 (BP).

Self-figure Drawing. The DAP is a well-documented psychological assessment tool that was first documented and utilized by [45] on a child population. This tool was later developed extensively by [46], who popularized the DAP as a personality assessment, and [47], who studied the utilization of the DAP to reflect self-image among children as adolescents. The purpose of analyzing self-figure drawing is to investigate further the issues a client is dealing with [32]. We used self-figure drawing to assess to what extent indicators that were previously found to indicate deterioration in well-being (such as facial expression [48], disconnected or shaking body line [49], and disconnection of arms and hands [32] would be found in these population, too.

Narrative. The narrative is inherently multidisciplinary and is an extension of the interpretive approaches in social sciences. Narrative lends itself to a qualitative inquiry to capture the rich data within stories. Surveys, questionnaires, and quantitative behavior analyses cannot capture the complexity of meaning embodied within stories [50]. Narrative analysis can record different viewpoints and interpret collected data to identify similarities and differences in experiences and actions. Stories are presumed to provide a holistic context that allows individuals to reflect on and reconstruct their personal, historical, and cultural experiences [50].

### 3.3. Data Analysis

Quantitative data. The Kolmogorov–Smirnoff test carried out for the normality tests. Univariate analysis was used to find the relation between drawing indicators (normal vs. not normal) and demographic and depression, and physical image measure variables. Continuous variables that followed normal distribution were analyzed by using a two-sample Student’s *t*-test and reported as the mean and the standard deviation. Continuous variables that did not follow normal distribution were analyzed by using the Mann–Whitney U test and reported as the median and the interquartile range. The Pearson chi-square test was used to analyze categorical variables.

The comparison of drawings indicators before and after diagnosis was performed by using the McNemar test for symmetry. A *p*-value greater than 0.05 indicates no significant change. An analysis was performed by SAS 9.4 version for windows. Alpha’s Cronbach and McDonald’s omega estimates were calculated by using the omega function of the psych package in R version 4.0.1. A *p*-value of 0.05 was considered to be significant.

*Qualitative data*. The drawings were encoded by using specific indicators that were found to differentiate between previous or current states. Indicators that relate to body image were included. For example, the omission of eyes indicates a sense of helplessness, depression, and anxiety; Shaking or disconnected body lines indicate anxiety and lack of control [35]. Four social workers evaluated the indicators in the two drawing sets. Indicators were included if there was a higher than 70% agreement between evaluators.

The narratives were encoded by using the following indicators: (1) narrative organization (restricted, flooded, and organized); (2) whether the narrative is dissociative (i.e., the narrative does not concentrate on the impact of the illness); (3) central theme in the narrative, as in the person’s fear/anxiety/lack of control, the person’s sadness/loneliness/body distortion, or no emotion; (4) the narrator’s emotional expression; and (5) the resolution/solution of the narrative (positive, negative, or neutral) and accordance between the narrative and the drawing (yes/no) [50].

## 4. Results

The Pearson correlation test was conducted between the study variables. Significant positive correlations were found between depression and the following body image factors: Appearance Orientation (AO) (extent of investment in one’s appearance) and Illness Orientation (extent of reactivity to being or becoming ill). In addition, significant positive correlations were found between the factors of mental health in Quality of life and Appearance Orientation, Fitness Orientation, and Illness Orientation.

Table 2 presents the measures distribution and alpha Cronbach and McDonald’s omega reliability measure. The measures that were found to be normally distributed are presented with mean and standard deviation. In contrast, the measures found to be not normally distributed are presented with median and inter-quartile ranges.

A 67-year-old male who is undergoing peritoneal dialysis narrated his self-figure drawings before being diagnosed and at present: “My body was once normally shaped, but after dialysis, there was a bloating in the abdomen, a kind of obesity. Suddenly I had something noticeable in my abdomen… it was always there. One cannot ignore it. At first, it affected me visually, later more and more physical limitations” (Figure 1).

A 75-year-old female narrated the two self-figured drawings: “I have difficulty walking; I get tired after a short walk. Following the treatments, I gained weight mainly in the abdomen. I try to maintain social gatherings but fail” (Figure 2).

A 70-year-old woman narrated her two self-figured drawings: “What affected me was not the treatment but the disease…I always need help; someone needs to bring my kids, help me…I think if I was not sick, I would go to work…I did not know anything, and I was just sick. They brought me to the hospital. I was told I had a kidney problem and did not agree to do dialysis at first” (Figure 3).

Table 3 and Table 4 present the drawing indicators before and after diagnosis.

### 4.1. Relation to Continuous Body Line (Disconnected vs. Connected)

A disconnected continuous body line before the diagnosis is not significantly associated with any variable. A disconnected continuous body line after diagnosis is associated with Lower Fitness Orientation (M = 2.3 vs. M = 2.9, *p* = 0.023). Among 66% of the sample, no change was apparent between the figures before and after diagnosis, *p* = 0.06.

#### Relation to Shaking Body Line (No vs. Yes)

No shaking body line before diagnosis was associated with Appearance Evaluation (M = 3.7 vs. M = 2.9, *p* = 0.042), Fitness Evaluation (M = 3.5 vs. M = 2.8, *p* = 0.039), Fitness Orientation (M = 3.0 vs. M = 2.3, *p* = 0.032), Health Evaluation (M = 2.7 vs. M = 2.1), and Physical Functioning (M = 38.2 vs. M = 23.1). A shaking body line after diagnosis was not significantly associated with any variable. A total of 76% of the sample did not change their body line following diagnosis, *p* = 0.76.

### 4.2. Relation to Face Expression (Sad vs. Happy)

A sad facial expression before diagnosis was associated with Lower Reported health (MED = 40 vs. MED = 60, *p* = 0.045). A sad facial expression after diagnosis was not significantly associated with any variable. A total of 66% of the sample had the same face expression (mainly sad) before and after diagnosis, *p* = 0.21.

### 4.3. Relation to Hands (Omitted vs. Connected)

Omitted hands before the diagnosis was associated with lower Illness Orientation (MED = 3.9 vs. MED = 4.6, *p* = 0.033), Physical Functioning (M = 24.3 vs. M = 42.9), and Fitness Evaluation (M = 2.9 vs. M = 3.7, *p* = 0.027). Omitted hands after diagnosis was associated with higher Appearance Evaluation (M = 3.3 vs. M = 2.1, *p* = 0.036). A total of 72% of the sample exhibited no change before and after diagnosis, *p* = 0.15.

### 4.4. Relation to Neck (Omitted vs. Connected)

No variable significantly characterized the omission of the neck before or after. A total of 86% of the sample exhibited no change before and after diagnosis.

### 4.5. Relation to Feet (No vs. Yes)

No feet in the drawing of before diagnosis was associated with lower physical limitation (MED = 0 vs. MED = 50, *p* = 0.024). No feet in the drawing of after diagnosis was associated with lower Illness Orientation (MED = 4.0 vs. MED = 4.6, *p* = 0.041). A total of 86% of the sample did not exhibit any change before and after the diagnosis, *p* = 1.00.

### 4.6. Relation to Gender Markers (No vs. Yes)

No gender markers in the drawing of before diagnosis was associated with fewer years of dialysis (MED = 2 vs. MED = 5) and lower physical limitation (MED = 0 vs. MED = 50, *p* = 0.017). No gender markers in the drawing of after diagnosis was associated with lower Appearance Orientation (M = 2.9 vs. M = 3.6, *p* = 0.036). A total of 79% of the sample did not exhibit any change before and after diagnosis, *p* = 0.41.

### 4.7. Relation to Eyes (No vs. Yes)

No eyes in the drawing of before diagnosis was associated with higher Appearance Orientation (M = 3.4 vs. M = 2.8, *p* = 0.018), Health Orientation (M = 4.0 vs. M = 3.3, *p* = 0.016), Illness Orientation (MED = 4.2 vs. MED = 2.9, *p* = 0.023), and Physical Functioning (M = 36.6 vs. M = 20.7, *p* = 0.025). No eyes after diagnosis were associated with higher Appearance Orientation (M = 3.4 vs. M = 2.7, *p* = 0.030), Health Orientation (M = 3.9 vs. M = 3.2, *p* = 0.038), and Illness Orientation (MED = 4.2 vs. MED = 2.8, *p* = 0.013). A total of 76% of the sample did not exhibit any change, *p* = 0.26.

### 4.8. Relation to X’s and Shadowing (No vs. Yes)

No for X’s and shadowing before and after dialysis were not significantly associated with any variable. A total of 76% of the sample did not exhibit a change before and after, *p* = 0.06.

### 4.9. Narratives Analysis

The phenomenological analysis of the narratives yielded three themes: The first was life despair; feelings of desperation expressed this theme—“sadness” and “I have no desire to continue living”. The second theme was the physical functional impairment. Participants referred to the inability to perform daily activities that were previously routine for them: “before the illness I was able to walk, travel... now I can’t even do groceries”, “I can’t work, needs a work without effort”, and “I was active... since dialysis, I cannot do anything”. Concerning their social life, participants stated the following: “Dialysis treatment ruined my social life, a glass of water, it sucks, I’m afraid to harm my body”, and “I was so active before since dialysis I cannot participate in any activity”. The third theme was body image. Participants referred to the significant changes in their physical appearance. For example, one said, “I had a normal weight, then I became fat because of the water in my abdomen, now I am thin because I do not eat”; “I handed over all the clothes, I look ugly”; “I was attractive once, no longer, I physically bodily feel uncomfortable”; and “my big stomach since the treatment bothers me a lot”.

## 5. Discussion

The current retrospective study sought to inquire into the issue of body image as reflected in drawings and narratives of individuals who suffer from ESRD and are treated by dialysis. Participants were asked to fill out a self-report questionnaire that measured their levels of depression, body image, and quality of life. They were asked to draw and narrate their self-figure before the diagnosis and currently. Upon completion, participants gave narratives to the two drawings. The results revealed high levels of depression and concerns regarding body fitness and weight. Concerning the drawings (prior and current), differences were found in body line and facial expression, but these were not found to be significant.

ESRD is a chronic life-changing disease. As such, it will likely impact patients’ views of their life situation. It is not surprising, therefore, that such life changes are experienced as a negative outcome and that sadness and worry are the main emotions. The high levels of depression and low quality of life found in the current study are consistent with studies focusing on other chronic diseases indicating that the way patients with chronic diseases experience their bodies alters [51]. Chronic disease patients may struggle with identity, self-esteem, a shrinking lifeworld, and a challenging, complicated reality [52]. The disease becomes the most influential part of the patients’ lives, affecting their physical health and functions, autonomy, freedom, and identity, or even has a potentially life-threatening component that threatens their life [53]. It is unsurprising, therefore, that patients with ESRD who are forced to change their lifestyle due to the dialysis treatment enter into a survival mode and exhibit high levels of depression.

The fact that the participants showed deep concerns about body fitness and physical weight was expected since ESRD affects patients’ daily lives practically and emotionally. The term “body image” describes one’s perception of the body. ESRD changes the physical body shape [54]. According to [42], fitness evaluation is the range of investment in being physically fit or athletically capable. High scorers value fitness and actively engage in activities to maintain their fitness. A low score refers to the value of physical fitness and does not regularly incorporate exercise activities into their lifestyles. The overweight preoccupation subscale assesses the construct reflecting fat anxiety, weight vigilance, weight loss, dieting, and eating restraint. In contrast to low scorers, high scorers are very concerned about becoming fat. Several studies have reported that obesity increased the risk for chronic kidney disease [55]. Ref. [56] reported, in a prospective study of Korean men without hypertension and diabetes, that weight gain was associated with the development of chronic kidney disease. This relationship was observed even in the normal-weight participants. The effect of weight change on incident chronic kidney disease was maintained irrespective of various potential confounders, including age, baseline GFR, BMI, uric acid, HDL cholesterol, and regular exercise [57]. In the current study population, weight gain may reflect non-adherence to fluid restriction instructions [58]. Therefore, it is unsurprising that participants in the present study were found to significantly worry about their body weight and fitness because it might be a sign of deterioration.

The fact that the self-figured drawings before and after the diagnosis did not differ significantly in most of the indicators, indicating depression, anxiety, and fatigue (disconnected, shaken body line, hollowed eyes, detached arms and hands, etc.) raises the question of whether participants felt ill long before they were diagnosed with ESRD. This may be consistent with chronic kidney disease’s low recognition [59].

Patients should be assessed annually to determine whether they are at an increased risk of chronic kidney disease based on clinical and sociodemographic factors. Another possible psychological explanation for the similarity between the drawings before and a long time after diagnosis could derive from the attempt to suppress good memories that might exacerbate despair. Suppression is the act of consciously suppressing one’s feelings, thoughts, and wills. Although evidence shows that recalling happy memories elicits positive emotions and enhances well-being [60], we wonder if, in the case of suffering from CKD in which there is no cure, positive memories of oneself add to one’s desperation rather than helping one cope.

### Limitations

This study has three significant limitations that seem to reduce the findings’ generalizability: The first limitation is the small convenient sample size and the fact that the quantitative questionnaire was filled out only in relation to the current state. Recruitment of the ESRD population is difficult due to the patients’ physical and emotional conditions. Second, drawing oneself figure retrospectively might be influenced by the current physical and emotional state, especially if it lasts long. Thus, depressed mood and physical exhaustion might jeopardize one’s ability to ruminate on previous healthy bodily states. Likewise, this would apply to the narratives given to each drawing. In addition, the fact that participants filled out the questionnaire while undergoing dialysis is likely to influence their emotional mood. It is noteworthy to underscore that the dialysis treatment affects the patients’ state of mind, that it might keep them in a survival state of mind that affects their ability to think about other issues that are not crucial for their survival. The third limitation derives from the fact that the calculations do not adjust for confounding factors such as age or comorbidities. These confounding factors might contribute significantly to the patients’ image of themselves, in addition to their ESRD diagnosis.

## 6. Conclusions and Practical Implications

The findings have some practical implications. Since these patients meet at the hospital every two days, locked in the same room for four hours, a support group can be developed. This might lower the feelings of loneliness and aloneness and give a voice to these patients. This is in line with a previous study that suggested that educational and supportive group therapy expands the interpersonal relationships of hemodialysis patients and affects the patient’s quality of life [61]. 

## Figures and Tables

**Figure 1 ijerph-19-10777-f001:**
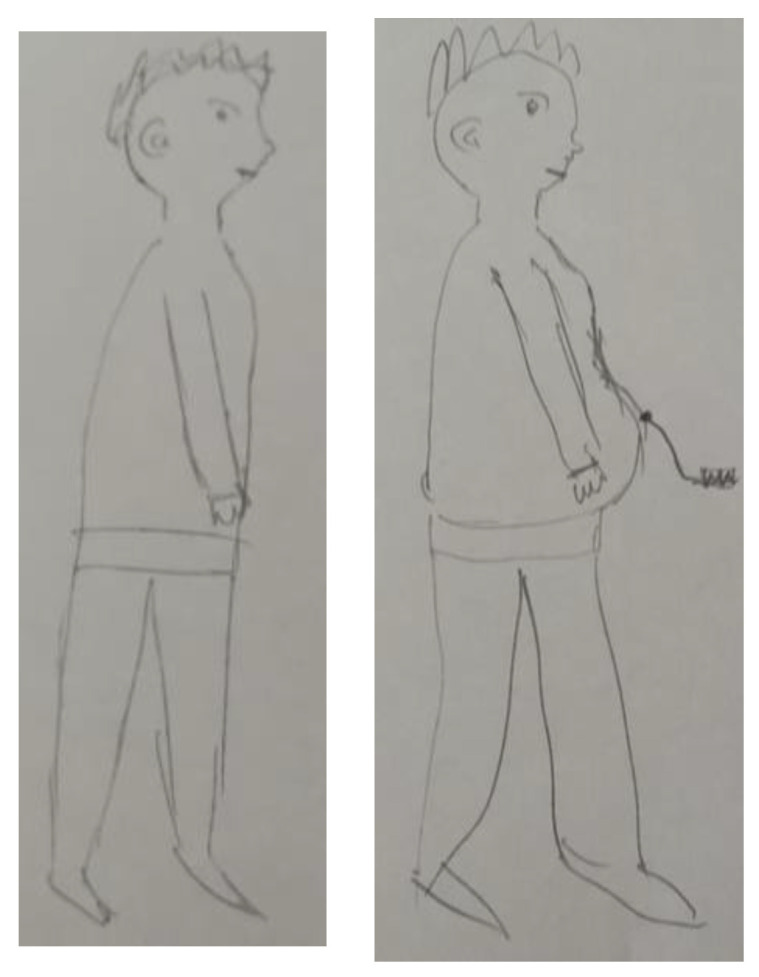
Self-figure drawing drawn by a 67-year-old male undergoing peritoneal dialysis: pre-diagnosis vs. at present.

**Figure 2 ijerph-19-10777-f002:**
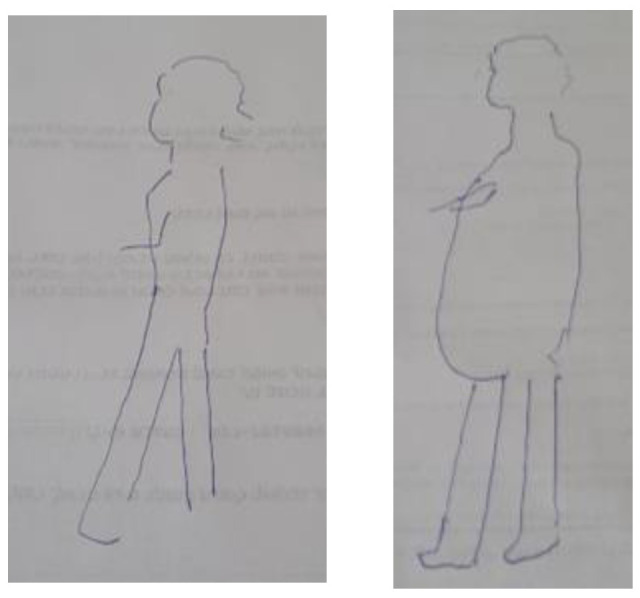
Self-figure drawing drawn by 75-year-old female undergoing hemodialysis or peritoneal dialysis: pre-diagnosis vs. at present.

**Figure 3 ijerph-19-10777-f003:**
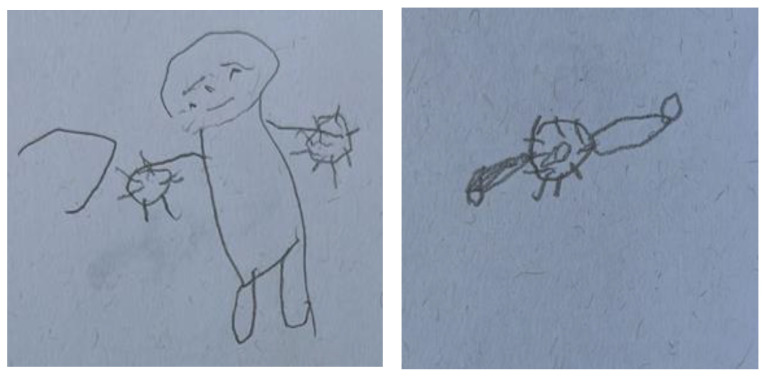
Self-figure drawing drawn by 70-year-old female undergoing hemodialysis or peritoneal dialysis: pre-diagnosis vs. at present.

**Table 1 ijerph-19-10777-t001:** Demographic variables distribution, N = 29.

Age, Median [25th;75th]	66.3 (9.06)
Gender, N (%):	
F	14 (48.3%)
M	15 (51.7%)
Family status, N (%):	
Married	15 (51.7%)
Not married	14 (48.3%)
Education, N (%):	
academic	4 (13.8%)
Non-academic	25 (86.2%)
Work status, N (%):	
(Unemployed) no	26 (89.7%)
(Employed) yes	3 (10.3%)
Years of dialysis, median [25th;75th]	3.00 [2.00;6.00]
Other than life-threatening diseases, N (%):	
no	4 (13.8%)
yes	25 (86.2%)

**Table 2 ijerph-19-10777-t002:** All scales distribution, N = 29.

Scale	MEAN (SD)/ Median [25th;75th]	(Min, Max)	Alpha	Omega
CES_D	17.1 (12.2)	(2.0, 40.0)	0.90	0.93
MBSQR Appearance Evaluation	3.20 (0.98)	(1.1, 5.0)	0.78	0.87
Appearance Orientation	3.11 (0.81)	(1.0, 4.4)	0.80	0.87
Fitness Evaluation	3.09 (0.87)	(1.0, 4.7)	0.22	0.31
Fitness Orientation	2.57 (0.77)	(1.0, 4.1)	0.76	0.83
Health Evaluation	2.30 (0.75)	(1.2, 4.3)	0.54	0.76
Health Orientation	3.66 (0.84)	(1.6, 5.0)	0.73	0.85
Illness Orientation	4.20 [2.80;4.60]	(1.4, 5.0)	0.79	0.91
Body Areas Satisfaction	3.46 (0.98)	(1.3, 5.0)	0.87	0.91
Overweight Preoccupation	2.78 (0.67)	(1.8, 4.0)	0.03	0.43
Self-Classified Weight	3.50 [3.00;4.00]	(1.0, 4.5)	0.83	–
Quality of Life SF-36 General Health	38.5 (22.9)	(5.0, 82.0)	0.68	0.86
Physical Functioning	28.8 (19.2)	(0, 65.0)	0.78	0.85
Role Limitation: Physical	0.00 [0.00;50.0]	(0, 100.0)	0.77	0.86
Role Limitation: Emotional	0.00 [0.00;66.7]	(0, 100.0)	0.87	0.88
Social Functioning	59.5 (35.9)	(0, 100.0)	0.85	–
Bodily pain	48.1 (32.2)	(0, 100.0)	0.92	–
Vitality	37.2 (23.6)	(0, 80.0)	0.80	0.88
Mental Health	62.6 (26.1)	(16.0, 96.0)	0.84	0.90
Reported Health	60.0 [40.0;80.0]	(0, 80.0)	–	

**Table 3 ijerph-19-10777-t003:** Drawing indicators for drawing of the self before diagnosis, N = 29.

Indicator	N (%)
Body line	
Connected	8 (27.6%)
Disconnected	21 (72.4%)
Shaking Body line	
No	18 (62.1%)
Yes	11 (37.9%)
Face expression	
Happy	16 (55.2%)
Sad	13 (44.8%)
Hands	
Connected	7 (24.1%)
Omitted	22 (75.9%)
Neck	
Connected	12 (41.4%)
Omitted	17 (58.6%)
Feet	
yes	8 (27.6%)
no	21 (72.4%)
Gender_Markers	
yes	11 (37.9%)
no	18 (62.1%)
Eyes1	
yes	14 (48.3%)
no	15 (51.7%)
X’s and shadowing	
no	26 (89.7%)
yes	3 (10.3%)

**Table 4 ijerph-19-10777-t004:** All indicators for drawing of the self after diagnosis, N = 29.

Indicator	N (%)
Body line	
Connected	14 (48.3%)
Disconnected	15 (51.7%)
Shaking Body line	
No	17 (58.6%)
Yes	12 (41.4%)
Face expression	
Happy	12 (41.4%)
Sad	17 (58.6%)
Hands	
Connected	3 (10.3%)
Omitted	26 (89.7%)
Neck	
Connected	12 (41.4%)
Omitted	17 (58.6%)
Feet	
yes	8 (27.6%)
no	21 (72.4%)
Gender_Markers	
yes	9 (31.0%)
no	20 (69.0%)
Eyes	
yes	11 (37.9%)
no	18 (62.1%)
X’s and shadowing	
no	21 (72.4%)
yes	8 (27.6%)

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
