# Peer review of "“Losing Faith in My Body”: Body Image in Individuals Diagnosed with End-Stage Renal Disease as Reflected in Drawings and Narratives"

_ijerph, 2022, doi:10.3390/ijerph191710777_

Round 1

Reviewer 1 Report

Congratulations to the authors for a nicely written paper on ESRD and its effect on patients image about themselves. The article is well written and easy to read and follow.

I have some observations before acceptance:

- What was the language of the self-reported questionnaire that was administered to the participants? Did the authors use translated questionnaires? (as the questionnaires used are originally validated but in english language). 

- No inclusion/exclusion criteria was presented.

- Some data in the Methods section would be more suitable for the Results section, such as Table 1.

- I identify one more important limitation. The calculations do not adjust for confounding factors such as age or comorbidities. These confounding factors may contribute significantly to the patients' image about themselves in addition to ESRD diagnosis.

- Regardind the last paragraph: "The findings have some practical implications. Since these patients meet at the hospital every two days, locked in the same room for four hours, a support group can be developed. This might lower the feelings of loneliness and aloneness and give voice to these patients." It would be interesting and impactful to include in the paper whether there are any published studies regarding the usefulnes and efficacy of such support groups (if they exist) in ESRD patients.

- A Conclusions section is missing.

Author Response

Dear Reviewer 1

Thanks for your comments, we appreciate the time and efforts you invested to help us improve the paper. Below are the revisions made:

  1. the questionnaires are originally developed in English but were adapted to Hebrew and were administered before in Hebrew-speaking populations
  2. Inclusion -exclusion criteria were added.
  3. Table 1 was placed in the Participants and Procedure section since it presents the demographics of the sample.
  4. The limitation suggested was added within the Discussion section.
  5. The Conclusions and limitations were separated into two sections.
  6. Please see the attachment

Reviewer 2 Report

Dear authors,

this study is really interesting as you give voice to end stage renal disease patients regarding their feelings, fears, worries and perceptions for their disease, life, body shape, physical function and social interaction. The information given in the present manuscript could be of great use to healthcare providers in order to support the patients emotionally, psychologically and physically.

My suggestions for improving the manuscript are:

1. Please give clearer results at the abstract (lines 17-19)

2. Table 1 should move to the results. Is age given in mean or median? (check between line 134 and the table)

3. line 152: the word originally does not need capital "O"

4. line 153: omitt "the CES-D''

5. line 163: change 10-1 and 16-11 to 1-10 and 11-16 respectively

6. line 237: it seems that a verb is missing

7. line 295: is "accordance between the narrative and the drawing" the 6th indicator?

8. In the first paragraph of your results you present the correlations you found. Have you adjusted the correlations with possible bias, i.e. age, years of dialysis, BMI, co-morbidities etc? It would reveal clearer correlations.

9. fig 1, 2 and 3 should come after a brief text/paragraph explaining what is shown bellow

10. Please connect patients' statements with the figures, ie I guess that statement in lines 313-316 comes from the same patient with the drawing shown in figure 1.

11. An introductory sentence for tables 3 and 4 will be helpful for the readers. Additionally, it would more convenient for comparison reasons to put the "Drawing indicators before and after diagnosis" in one table (Table 3). You could also provide p values for statistically significant differences.

12. line 421. Please check the sentence, it seems that a word is missing

13. line 439. The meaning of the sentence is not clear. Is "recognition low" the other way round (low recognition)? Do you may mean "underdiagnosed"?

14. I think that the practical implications can be enriched with some more ideas.

Author Response

Dear Reviewer 2,

We appreciate the time and effort you invested in reviewing the paper.

The following are the revisions made according to your comments:

  1. The Results in the Abstract were clarified.
  2. Table 1 remained in the Method below the Participants and Procedure section since it presents the sample demographics. 
  3. comments 3-7 were changed as suggested.
  4. The purpose of testing the correlations was just to show a relationship between depression and body image in this specific situation. We did not mean to characterize depression or body image of the quality of life by demographic factors. 
  5. the narratives were paced before the Figures as suggested.
  6. An introductory statement for Tables was added.
  7. Low recognition means underdiagnosed. It was clarified.
  8. The Implication section was elaborated as suggested.
  9. Please see the attachment

Reviewer 3 Report

In this article, the authors investigated how individuals who undergo chronic dialysis due to failure end-stage renal disease 9 perceive their bodies as reflected in drawings and narratives. Researchers enrolled 29 CKD-patients (females = 14; males = 15), aged 20 to 85 years, treated by chronic hemodialysis or peritoneal dialysis (from 1 years to 27 years, median = 3 years). After signing the consent form, participants completed anonymous self-assessment questionnaires: (CES-D), (MBSRQ) and (SF-36). After completion, they were asked to draw their own self-image before and after the diagnosis and to tell about it. Unfortunately, from my point of view, there are several methodological aspects that need to be improved:

- The authors did not specify what the nature of the study I can imagine is an observational study (retrospective study? Prospective study? Longitudinal study?);

- the sample size is very small

- The authors did not specify the timing of delivery of the questionnaires;

- (CES-D), (MBSRQ) and (SF-36) questionnaires were administered only once while the authors asked the participants to draw their own self-shape before and after the dialysis period;

- In Tab1 the descriptive variables are poor but in any case there is important information that the authors had not studied in depth (for example, "Other diseases", yes= 86.2% or "Work status", no= 89.7% or "Education”, non-acedemic=86.2%). These major imbalances could be a confounding factor.

- In Tab 2 the data are expressed in mean (sd) or median (IQR) but it is important to add another column with range score (max, min). Furthermore, the Alpha coefficient alone does not have much value and such high or low values ​​(0.9,0.92,0.87 or 0.02) are generally unlikely. Alpha is conditioned by the number of items and the sample size. In this case the articles are relatively many items and the sample is too small. Authors should reevaluate Alpha for example with a correlation matrix or to use a more sensitive Omega coefficient.

- In Tab 3 and 4 the authors presented the drawings indicators before and after diagnosis but they not did evaluate the possible differences between these indicators before and after with relative p-value.

- The authors did not perform a gender stratified analysis or for hemodialysis and peritoneal treatment.

Author Response

Dear Reviewer 3'

Thank you for investing time and effort in this paper. we have made the following revisions according to your comments:

  1. This is a retrospective study as you commented. It was mentioned as such at the end of the Introduction, the Method, and the beginning of the Discussion.
  2. The small sample size was mentioned as a limitation.
  3. The timing of filling out the questionnaire was added in the Procedure section.
  4. The fact that the quantitative measures were administered only in relation to the present was mentioned as a limitation. However, this was done because we learned after the two first interviews we conducted, that the patients found it too difficult to relate to their previous state, it aggravated them to the point of resisting participation and exhausted them. 
  5. The meaning of employment status and other diseases were added in Table 1. 
  6. Alpha was re-evaluated as suggested and an additional column was added to the Table.
  7. Max and min values were added to table 2 as well as the Omega coefficient., which turned out to be higher than the alpha in all scales. In the explanation below in Tables 3 and 4, we relate to differences between these indicators before and after, we write there that for most of the participants we found no difference in any indicator. We added the p-value of the Mc'nemar test that tested the difference between drawings indicators before and after diagnosis.
  8. We found no significant difference across genders regarding drawing indicators and scales and therefore we haven't related to that in the text.
  9. Please see the attachment

Round 2

Reviewer 3 Report

The authors proceeded to modify and improve the methods and the statistical analysis.